# Multilinear Dynamical Systems for Tensor Time Series

**Mark Rogers**      **Lei Li**      **Stuart Russell**
EECS Department, University of California, Berkeley
markrogersjr@berkeley.edu, {leili,russell}@cs.berkeley.edu

## Abstract

Data in the sciences frequently occur as sequences of multidimensional arrays called tensors. How can hidden, evolving trends in such data be extracted while preserving the tensor structure? The model that is traditionally used is the linear dynamical system (LDS) with Gaussian noise, which treats the latent state and observation at each time slice as a vector. We present the multilinear dynamical system (MLDS) for modeling tensor time series and an expectation–maximization (EM) algorithm to estimate the parameters. The MLDS models each tensor observation in the time series as the multilinear projection of the corresponding member of a sequence of latent tensors. The latent tensors are again evolving with respect to a multilinear projection. Compared to the LDS with an equal number of parameters, the MLDS achieves higher prediction accuracy and marginal likelihood for both artificial and real datasets.

## 1  Introduction

A tenet of mathematical modeling is to faithfully match the structural properties of the data; yet, on occasion, the available tools are inadequate to perform the task. This scenario is especially common when the data are tensors, i.e., multidimensional arrays: vector and matrix models are fitted to them without justification. This is, perhaps, due to the lack of an agreed-upon tensor model. There are many examples that seem to require such a model: The spatiotemporal grid of atmospheric data in climate modeling is a time series of $n \times m \times l$ tensors, where $n$, $m$ and $l$ are the numbers of latitude, longitude, and elevation grid points. If $k$ measurements—e.g., temperature, humidity, and wind speed for $k=3$—are made, then a time series of $n \times m \times l \times k$ tensors is constructed. The daily high, low, opening, closing, adjusted closing, and volume of the stock prices of $n$ multiple companies comprise a time series of $6 \times n$ tensors. A grayscale video sequence is a two-dimensional tensor time series because each frame is a two-dimensional array of pixels.

Several queries can be made when one is presented with a tensor time series. As with any time series, a forecast of future data may be requested. For climate data, successful prediction may spell out whether the overall ocean temperatures will increase. Prediction of stock prices may not only inform investors but also help to stabilize the economy and prevent market collapse. The relationships between particular subsets of tensor elements could be of significance. How does the temperature of the ocean at 8°N, 165°E affect the temperature at 5°S, 125°W? For stock price data, one may investigate how the stock prices of electric car companies affect those of oil companies. For a video sequence, one might expect adjacent pixels to be more correlated than those far away from each other. Another way to describe the relationships among tensor elements is in terms of their covariances. Equipped with a tabulation of the covariances, one may read off how a given tensor element affects others. Later in this paper, we will define a tensor time series model and a covariance tensor that permits the modeling of general noise relationships among tensor elements.

More formally, a *tensor* $\mathcal{X} \in \mathbf{R}^{I_1 \times \cdots \times I_M}$ is a multidimensional array with elements that can each be indexed by a vector of positive integers. That is, every element $\mathcal{X}_{i_1 \cdots i_M} \in \mathbf{R}$ is uniquely addressed

by a vector $(i_1, \cdots, i_M)$ such that $1 \leq i_m \leq I_m$ for all $m$. Each of the $M$ dimensions of $\mathfrak{X}$ is called a *mode* and represents a particular component of the data. The simplest tensors are vectors and matrices: vectors are tensors with only a single mode, while matrices are tensors with two modes. We will consider the *tensor time series*, which is an ordered, finite collection of tensors that all share the same dimensionality. In practice, each member of an observed tensor time series reflects the state of a dynamical system that is measured at discrete epochs.

We propose a novel model for tensor time series: the multilinear dynamical system (MLDS). The MLDS explicitly incorporates the dynamics, noise, and tensor structure of the data by juxtaposing concepts in probabilistic graphical models and multilinear algebra. Specifically, the MLDS generalizes the states of the linear dynamical system (LDS) to tensors via a probabilistic variant of the Tucker decomposition. The LDS tracks latent vector states and observed vector sequences; this permits forecasting, estimation of latent states, and modeling of noise but only for vector objects. Meanwhile, the Tucker decomposition of a single tensor computes a latent "core" tensor but has no dynamics or noise capabilities. Thus, the MLDS achieves the best of both worlds by uniting the two models in a common framework. We show that the MLDS, in fact, generalizes LDS and other well-known vector models to tensors of arbitrary dimensionality. In our experiments on both synthetic and real data, we demonstrate that the MLDS outperforms the LDS with an equal number of parameters.

## 2  Tensor algebra

Let $\mathbf{N}$ be the set of all positive integers and $\mathbf{R}$ be the set of all real numbers. Given $I \in \mathbf{N}^M$, where $M \in \mathbf{N}$, we assemble a *tensor-product space* $\mathbf{R}^{I_1 \times \cdots \times I_M}$, which will sometimes be written as $\mathbf{R}^I = \mathbf{R}^{(I_1, \ldots, I_M)}$ for shorthand. Then a *tensor* $\mathfrak{X} \in \mathbf{R}^{I_1 \times \cdots \times I_M}$ is an element of a tensor-product space. A tensor $\mathfrak{X}$ may be referenced by either a full vector $(i_1, \ldots, i_M)$ or a by subvector, using the $\bullet$ symbol to indicate coordinates that are not fixed. For example, let $\mathfrak{X} \in \mathbf{R}^{I_1 \times I_2 \times I_3}$. Then $\mathfrak{X}_{i_1 i_2 i_3}$ is a scalar, $\mathfrak{X}_{\bullet i_2 i_3} \in \mathbf{R}^{I_1}$ is the vector obtained by setting the second and third coordinates to $i_2$ and $i_3$, and $\mathfrak{X}_{\bullet \bullet i_3} \in \mathbf{R}^{I_1 \times I_2}$ is the matrix obtained by setting the third coordinate to $i_3$. The concatenation of two $M$-dimensional vectors $I = (I_1, \ldots, I_M)$ and $J = (J_1, \ldots, J_M)$ is given by $IJ = (I_1, \ldots, I_M, J_1, \ldots, J_M)$, a vector with $2M$ entries.

Let $\mathfrak{X} \in \mathbf{R}^{I_1 \times \cdots \times I_M}$, $M \in \mathbf{N}$. The *vectorization* $\mathrm{vec}(\mathfrak{X}) \in \mathbf{R}^{I_1 \cdots I_M}$ is obtained by shaping the tensor into a vector. In particular, the elements of $\mathrm{vec}(\mathfrak{X})$ are given by $\mathrm{vec}(\mathfrak{X})_k = \mathfrak{X}_{i_1 \cdots i_M}$, where $k = 1 + \sum_{m=1}^{M} \prod_{n=1}^{m-1} I_n (i_m - 1)$. For example, if $\mathfrak{X} \in \mathbf{R}^{2 \times 3 \times 2}$ is given by

$$\mathfrak{X}_{\bullet \bullet 1} = \begin{pmatrix} 1 & 3 & 5 \\ 2 & 4 & 6 \end{pmatrix}, \quad \mathfrak{X}_{\bullet \bullet 2} = \begin{pmatrix} 7 & 9 & 11 \\ 8 & 10 & 12 \end{pmatrix},$$

then $\mathrm{vec}(\mathfrak{X}) = (1\ 2\ 3\ 4\ 5\ 6\ 7\ 8\ 9\ 10\ 11\ 12)^{\mathrm{T}}$.

Let $I, J \in \mathbf{N}^M, M \in \mathbf{N}$. The *matricization* $\mathrm{mat}(\mathcal{A}) \in \mathbf{R}^{I_1 \cdots I_M \times J_1 \cdots J_M}$ of a tensor $\mathcal{A} \in \mathbf{R}^{IJ}$ is given by $\mathrm{mat}(\mathcal{A})_{kl} = \mathcal{A}_{i_1 \cdots i_M j_1 \cdots j_M}$, where $k = 1 + \sum_{m=1}^{M} \prod_{n=1}^{m-1} I_n (i_m - 1)$ and $l = 1 + \sum_{m=1}^{M} \prod_{n=1}^{m-1} J_n (j_m - 1)$. The matricization "flattens" a tensor into a matrix. For example, define $\mathcal{A} \in \mathbf{R}^{2 \times 2 \times 2 \times 2}$ by

$$\mathcal{A}_{\bullet \bullet 11} = \begin{pmatrix} 1 & 3 \\ 2 & 4 \end{pmatrix}, \mathcal{A}_{\bullet \bullet 21} = \begin{pmatrix} 5 & 7 \\ 6 & 8 \end{pmatrix}, \mathcal{A}_{\bullet \bullet 12} = \begin{pmatrix} 9 & 11 \\ 10 & 12 \end{pmatrix}, \mathcal{A}_{\bullet \bullet 22} = \begin{pmatrix} 13 & 15 \\ 14 & 16 \end{pmatrix}.$$

Then we have $\mathrm{mat}(\mathcal{A}) = \begin{pmatrix} 1 & 5 & 9 & 13 \\ 2 & 6 & 10 & 14 \\ 3 & 7 & 11 & 15 \\ 4 & 8 & 12 & 16 \end{pmatrix}$.

The vec and mat operators put tensors in bijective correspondence with vectors and matrices. To define the inverse of each of these operators, a reference must be made to the dimensionality of the original tensor. In other words, given $\mathfrak{X} \in \mathbf{R}^I$ and $\mathcal{A} \in \mathbf{R}^{IJ}$, where $I, J \in \mathbf{N}^M, M \in \mathbf{N}$, we have $\mathfrak{X} = \mathrm{vec}_I^{-1}(\mathrm{vec}(\mathfrak{X}))$ and $\mathcal{A} = \mathrm{mat}_{IJ}^{-1}(\mathrm{mat}(\mathcal{A}))$.

Let $I, J \in \mathbf{N}^M, M \in \mathbf{N}$. The *factorization* of a tensor $\mathcal{A} \in \mathbf{R}^{IJ}$ is given by $\mathcal{A}_{i_1 \cdots i_M j_1 \cdots j_M} = \prod_{m=1}^{M} A_{i_m j_m}^{(m)}$, where $A^{(m)} \in \mathbf{R}^{I_m \times J_m}$ for all $m$. The factorization exponentially reduces the

number of parameters needed to express $\mathcal{A}$ from $\prod_{m=1}^{M} I_m J_m$ to $\sum_{m=1}^{M} I_m J_m$. In matrix form, we have $\mathrm{mat}(\mathcal{A}) = A^{(M)} \otimes A^{(M-1)} \otimes \cdots \otimes A^{(1)}$, where $\otimes$ is the Kronecker matrix product [1]. Note that tensors in $\mathbf{R}^{IJ}$ are not factorizable in general [2].

The *product* $\mathcal{A} \circledast \mathcal{X}$ of two tensors $\mathcal{A} \in \mathbf{R}^{IJ}$ and $\mathcal{X} \in \mathbf{R}^{J}$, where $I, J \in \mathbf{N}^M, M \in \mathbf{N}$, is given by $(\mathcal{A} \circledast \mathcal{X})_{i_1 \cdots i_M} = \sum_{j_1 \cdots j_M} \mathcal{A}_{i_1 \cdots i_M j_1 \cdots j_M} \mathcal{X}_{j_1 \cdots j_M}$. The tensor $\mathcal{A}$ is called a *multilinear operator* when it appears in a tensor product as above. The product is only defined if the dimensionalities of the last $M$ modes of $\mathcal{A}$ match the dimensionalities of $\mathcal{X}$. Note that this tensor product generalizes the standard matrix-vector product in the case $M = 1$.

We shall primarily work with tensors in their vector and matrix representations. Hence, we appeal to the following

**Lemma 1.** *Let* $I, J \in \mathbf{N}^M, M \in \mathbf{N}, \mathcal{A} \in \mathbf{R}^{IJ}, \mathcal{X} \in \mathbf{R}^{J}$. *Then*

$$\mathrm{vec}(\mathcal{A} \circledast \mathcal{X}) = \mathrm{mat}(\mathcal{A}) \, \mathrm{vec}(\mathcal{X}) . \tag{1}$$

*Furthermore, if $\mathcal{A}$ is factorizable with matrices $A^{(m)}$, then*

$$\mathrm{vec}(\mathcal{A} \circledast \mathcal{X}) = \left[ A^{(M)} \otimes \cdots \otimes A^{(1)} \right] \mathrm{vec}(\mathcal{X}) . \tag{2}$$

*Proof.* Let $k = 1 + \sum_{m=1}^{M} \prod_{n=1}^{m-1} I_n(i_m - 1)$ and $l = 1 + \sum_{m=1}^{M} \prod_{n=1}^{m-1} J_n(j_m - 1)$ for some $(j_1, \ldots, j_M)$. We have

$$\mathrm{vec}(\mathcal{A} \circledast \mathcal{X})_k = \sum_{j_1 \cdots j_M} \mathcal{A}_{i_1 \cdots i_M j_1 \cdots j_M} \mathcal{X}_{j_1 \cdots j_M} = \sum_{l} \mathrm{mat}(\mathcal{A})_{kl} \, \mathrm{vec}(\mathcal{X})_l = (\mathrm{mat}(\mathcal{A}) \, \mathrm{vec}(\mathcal{X}))_k ,$$

which holds for all $1 \le i_m \le I_m, 1 \le m \le M$. Thus, (1) holds. To prove (2), we express $\mathrm{mat}(\mathcal{A})$ as the Kronecker product of $M$ matrices $A^{(1)}, \ldots, A^{(M)}$. $\square$

The Tucker decomposition can be expressed using the product $\circledast$ defined above. The Tucker decomposition models a given tensor $\mathcal{X} \in \mathbf{R}^{I_1 \times \cdots \times I_M}$ as the result of a multilinear transformation that is applied to a latent core tensor $\mathcal{Z} \in \mathbf{R}^{J_1 \times \cdots \times J_M}$: $\mathcal{X} = \mathcal{A} \circledast \mathcal{Z}$.

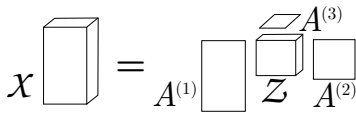

The multilinear operator $\mathcal{A}$ is a factorizable tensor such that $\mathrm{mat}(\mathcal{A}) = A^{(M)} \otimes A^{(M-1)} \otimes \cdots \otimes A^{(1)}$, where $A^{(1)}, \ldots, A^{(M)}$ are *projection matrices* (Figure 1). The canonical decomposition/parallel factors (CP) decomposition is a special case of the Tucker decomposition in which $\mathcal{Z}$ is "superdiagonal", i.e., $J_1 = \cdots = J_M = R$ and only the $\mathcal{Z}_{j_1 \cdots j_M}$ such that $j_1 = \cdots = j_M$ can be nonzero. The CP decomposition expresses $\mathcal{X}$ as a sum

**Figure 1:** The Tucker decomposition of a third-order tensor $\mathcal{X}$.

$\mathcal{X} = \sum_{r=1}^{R} u_r^{(1)} \circ \cdots \circ u_r^{(M)}$, where $u_r^{(m)} \in \mathbf{R}^{I_m}$ for all $m$ and $r$ and $\circ$ denotes the tensor outer product [3].

To illustrate, consider the case $M = 2$ and let $X = \mathcal{A} \circledast Z$, where $X \in \mathbf{R}^{n \times m}$ and $Z \in \mathbf{R}^{p \times q}$. Then $X = AZB^{\mathrm{T}}$, where $\mathrm{mat}(\mathcal{A}) = B \otimes A$. If $p \le n$ and $q \le m$, then $Z$ is a dimensionality-reduced version of $X$: the matrix $A$ increases the number of rows of $Z$ from $p$ to $n$ via left-multiplication, while the matrix $B$ increases the number of columns of $Z$ from $q$ to $m$ via right-multiplication. To reconstruct $X$, we simply apply $\mathcal{A} \circledast Z$. See Figure 1 for an illustration of the case $M = 3$.

## 3 Random tensors

Given $I \in \mathbf{N}^M, M \in \mathbf{N}$, we define a random tensor $\mathcal{X} \in \mathbf{R}^{I_1 \times \cdots \times I_M}$ as follows. Suppose $\mathrm{vec}(\mathcal{X})$ is normally distributed with expectation $\mathrm{vec}(\mathcal{U})$ and positive-definite covariance $\mathrm{mat}(\mathcal{S})$, where $\mathcal{U} \in \mathbf{R}^I$ and $\mathcal{S} \in \mathbf{R}^{II}$. Then we say that $\mathcal{X}$ has the *normal distribution* with expectation $\mathcal{U} \in \mathbf{R}^I$ and covariance $\mathcal{S} \in \mathbf{R}^{II}$ and write $\mathcal{X} \sim \mathcal{N}(\mathcal{U}, \mathcal{S})$. The definition of the normal distribution on tensors can thus be restated more succinctly as

$$\mathcal{X} \sim \mathcal{N}(\mathcal{U}, \mathcal{S}) \iff \mathrm{vec}(\mathcal{X}) \sim \mathcal{N}(\mathrm{vec}\, \mathcal{U}, \mathrm{mat}\, \mathcal{S}) . \tag{3}$$

Our formulation extends the normal distribution defined in [4], which is restricted to symmetric, second-order tensors.

We will make use of an important special case of the normal distribution defined on tensors: the *multilinear Gaussian distribution*. Let $I, J \in \mathbf{N}^M, M \in \mathbf{N}$, and suppose $\mathcal{X} \in \mathbf{R}^I$ and $\mathcal{Z} \in \mathbf{R}^J$ are jointly distributed as

$$\mathcal{Z} \sim \mathcal{N}\left(\mathcal{U}, \mathcal{G}\right) \text{ and } \mathcal{X} \mid \mathcal{Z} \sim \mathcal{N}\left(\mathcal{C} \circledast \mathcal{Z}, \mathcal{S}\right), \tag{4}$$

where $\mathcal{C} \in \mathbf{R}^{IJ}$. The marginal distribution of $\mathcal{X}$ and the posterior distribution of $\mathcal{Z}$ given $\mathcal{X}$ are given by the following result.

**Lemma 2.** *Let $I, J \in \mathbf{N}^M, M \in \mathbf{N}$, and suppose the joint distribution of random tensors $\mathcal{X} \in \mathbf{R}^I$ and $\mathcal{Z} \in \mathbf{R}^J$ is given by* (4)*. Then the marginal distribution of $\mathcal{X}$ is*

$$\mathcal{X} \sim \mathcal{N}\left(\mathcal{C} \circledast \mathcal{U}, \mathcal{C} \circledast \mathcal{G} \circledast \mathcal{C}^{\mathsf{T}} + \mathcal{S}\right), \tag{5}$$

*where $\mathcal{C}^{\mathsf{T}} \in \mathbf{R}^{JI}$ and $\mathcal{C}^{\mathsf{T}}_{j_1 \cdots j_M i_1 \cdots i_M} = \mathcal{C}_{i_1 \cdots i_M j_1 \cdots j_M}$. The conditional distribution of $\mathcal{Z}$ given $\mathcal{X}$ is*

$$\mathcal{Z} \mid \mathcal{X} \sim \mathcal{N}\left(\hat{\mathcal{U}}, \hat{\mathcal{G}}\right), \tag{6}$$

*where $\hat{\mathcal{U}} = \mathrm{vec}_J^{-1}\left(\mu + W\left(\mathrm{vec}(\mathcal{X}) - \mathrm{mat}(\mathcal{C})\,\mu\right)\right)$, $\hat{\mathcal{G}} = \mathrm{mat}_{JJ}^{-1}\left(\Gamma - W\mathrm{mat}(\mathcal{C})\,\Gamma\right)$, $\mu = \mathrm{vec}(\mathcal{U})$, $\Gamma = \mathrm{mat}(\mathcal{G})$, $\Sigma = \mathrm{mat}(\mathcal{S})$, and $W = \Gamma\mathrm{mat}(\mathcal{C})^{\mathsf{T}}\left[\mathrm{mat}(\mathcal{C})\,\Gamma\mathrm{mat}(\mathcal{C})^{\mathsf{T}} + \Sigma\right]^{-1}$.*

*Proof.* Lemma 1, (3), and (4) imply that the vectorizations of $\mathcal{Z}$ and $\mathcal{X}$ given $\mathcal{Z}$ follow $\mathrm{vec}(\mathcal{Z}) \sim \mathcal{N}\left(\mu, \Gamma\right)$ and $\mathrm{vec}(\mathcal{X}) \mid \mathrm{vec}(\mathcal{Z}) \sim \mathcal{N}\left(\mathrm{mat}(\mathcal{C})\,\mathrm{vec}(\mathcal{Z}), \Sigma\right)$. By the properties of the multivariate normal distribution, the marginal distribution of $\mathrm{vec}(\mathcal{X})$ and the conditional distribution of $\mathrm{vec}(\mathcal{Z})$ given $\mathrm{vec}(\mathcal{X})$ are $\mathrm{vec}(\mathcal{X}) \sim \mathcal{N}(\mathrm{mat}(\mathcal{C})\,\mathrm{vec}(\mathcal{U}), \mathrm{mat}(\mathcal{C})\,\Gamma\mathrm{mat}(\mathcal{C})^{\mathsf{T}} + \Sigma)$ and $\mathrm{vec}(\mathcal{Z}) \mid \mathrm{vec}(\mathcal{X}) \sim \mathcal{N}(\mathrm{vec}(\hat{\mathcal{U}}), \mathrm{mat}(\hat{\mathcal{G}}))$. The associativity of $\circledast$ implies that $\mathrm{mat}(\mathcal{C} \circledast \mathcal{G} \circledast \mathcal{C}^{\mathsf{T}}) = \mathrm{mat}(\mathcal{C})\,\Gamma\mathrm{mat}(\mathcal{C})^{\mathsf{T}}$. Finally, we apply Lemma 1 once more to obtain (5) and (6). $\square$

## 4 Multilinear dynamical system

The aim is to develop a model of a tensor time series $\mathcal{X}_1, \ldots, \mathcal{X}_N$ that takes into account tensor structure. In defining the MLDS, we build upon the results of previous sections by treating each $\mathcal{X}_n$ as a random tensor and relating the model components with multilinear transformations. When the MLDS components are vectorized and matricized, an LDS with factorized transition and projection matrices is revealed. Hence, the strategy for fitting the MLDS is to vectorize each $\mathcal{X}_n$, run the expectation-maximization (EM) algorithm of the LDS for all components but the matricized transition and projection tensors–which are learned via an alternative gradient method–and finally convert all model components back to tensor form.

### 4.1 Definition

Let $I, J \in \mathbf{N}^M, M \in \mathbf{N}$. The MLDS model consists of a sequence $\mathcal{Z}_1, \ldots, \mathcal{Z}_N$ of latent tensors, where $\mathcal{Z}_n \in \mathbf{R}^{J_1 \times \cdots \times J_M}$ for all $n$. Each latent tensor $\mathcal{Z}_n$ emits an observation $\mathcal{X}_n \in \mathbf{R}^{I_1 \times \cdots \times I_M}$. The system is initialized by a latent tensor $\mathcal{Z}_1$ distributed as

$$\mathcal{Z}_1 \sim \mathcal{N}\left(\mathcal{U}_0, \mathcal{Q}_0\right). \tag{7}$$

Given $\mathcal{Z}_n, 1 \leq n \leq N-1$, we generate $\mathcal{Z}_{n+1}$ according to the conditional distribution

$$\mathcal{Z}_{n+1} \mid \mathcal{Z}_n \sim \mathcal{N}\left(\mathcal{A} \circledast \mathcal{Z}_n, \mathcal{Q}\right), \tag{8}$$

where $\mathcal{Q}$ is the conditional covariance shared by all $\mathcal{Z}_n, 2 \leq n \leq N$, and $\mathcal{A}$ is the *transition tensor* which describes the dynamics of the evolving sequence $\mathcal{Z}_1, \ldots, \mathcal{Z}_N$. The transition tensor $\mathcal{A}$ is factorized into $M$ matrices $A^{(m)}$, each of which acts on a mode of $\mathcal{Z}_n$. In matrix form, we have $\mathrm{mat}(\mathcal{A}) = A^{(M)} \otimes \cdots \otimes A^{(1)}$. To each $\mathcal{Z}_n$ there corresponds an observation $\mathcal{X}_n$ generated by

$$\mathcal{X}_n \mid \mathcal{Z}_n \sim \mathcal{N}\left(\mathcal{C} \circledast \mathcal{Z}_n, \mathcal{R}\right), \tag{9}$$

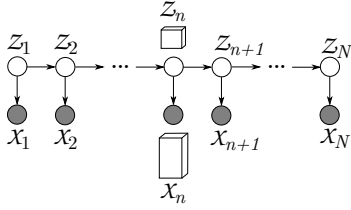

**Figure 2:** Schematic of the MLDS with three modes.

where $\mathcal{R}$ is the covariance shared by all $\mathcal{X}_n$ and $\mathcal{C}$ is the *projection tensor* which multilinearly transforms the latent tensor $\mathcal{Z}_n$. Like the transition tensor $\mathcal{A}$, the projection tensor $\mathcal{C}$ is factorizable, i.e., $\mathrm{mat}(\mathcal{C}) = C^{(M)} \otimes \cdots \otimes C^{(1)}$. See Figure 2 for an illustration of the MLDS.

By vectorizing each $\mathcal{X}_n$ and $\mathcal{Z}_n$, the MLDS becomes an LDS with factorized transition and projection matrices $\mathrm{mat}(\mathcal{A})$ and $\mathrm{mat}(\mathcal{C})$. For the LDS, the transition and projection operators are not factorizable in general [2]. The factorizations of $\mathcal{A}$ and $\mathcal{C}$ for the MLDS not only allow for a generalized dimensionality reduction of tensors but exponentially reduce the number of parameters of the transition and projection operators from $|A^{\mathrm{LDS}}| + |C^{\mathrm{LDS}}| = \prod_{m=1}^{M} J_m^2 + \prod_{m=1}^{M} I_m J_m$ down to $|\mathcal{A}^{\mathrm{MLDS}}| + |\mathcal{C}^{\mathrm{MLDS}}| = \sum_{m=1}^{M} J_m^2 + \sum_{m=1}^{M} I_m J_m$.

## 4.2 Parameter estimation

Given a sequence of observations $\mathcal{X}_1, \ldots, \mathcal{X}_N$, we wish to fit the MLDS model by estimating $\theta = (\mathcal{U}_0, \mathcal{Q}_0, \mathcal{Q}, \mathcal{A}, \mathcal{R}, \mathcal{C})$. Because the MLDS model contains latent variables $\mathcal{Z}_n$, we cannot directly maximize the likelihood of the data with respect to $\theta$. The EM algorithm circumvents this difficulty by iteratively updating $(\mathrm{E}(\mathcal{Z}_1), \ldots, \mathrm{E}(\mathcal{Z}_N))$ and $\theta$ in an alternating manner until the expected, complete likelihood of the data converges [5]. The normal distribution of tensors (3) will facilitate matrix and vector computations rather than compel us to work directly with tensors. In particular, we can express the complete likelihood of the MLDS model as

$$L\left(\theta \mid \mathcal{Z}_1, \mathcal{X}_1, \ldots, \mathcal{Z}_N, \mathcal{X}_N\right) = L\left(\mathrm{vec}\,\theta \mid \mathrm{vec}\,\mathcal{Z}_1, \mathrm{vec}\,\mathcal{X}_1, \ldots, \mathrm{vec}\,\mathcal{Z}_N, \mathrm{vec}\,\mathcal{X}_N\right), \quad (10)$$

where $\mathrm{vec}\,\theta = (\mathrm{vec}\,\mathcal{U}_0, \mathrm{mat}\,\mathcal{Q}_0, \mathrm{mat}\,\mathcal{Q}, \mathrm{mat}\,\mathcal{A}, \mathrm{mat}\,\mathcal{R}, \mathrm{mat}\,\mathcal{C})$. It follows that the vectorized MLDS is an LDS that inherits the Kalman filter updates for the E-step and the M-step for all parameters except $\mathrm{mat}\,\mathcal{A}$ and $\mathrm{mat}\,\mathcal{C}$. See [6] for the EM algorithm of the LDS.

Because $\mathcal{A}$ and $\mathcal{C}$ are factorizable, an alternative to the standard LDS updates is required. We locally maximize the expected, complete log-likelihood by computing the gradient with respect to the vector $v = [\mathrm{vec}\,C^{(1)\mathrm{T}} \cdots \mathrm{vec}\,C^{(M)\mathrm{T}}]^{\mathrm{T}} \in \mathbf{R}^{\sum_m I_m J_m}$, which is obtained by concatenating the vectorizations of the projection matrices $C^{(m)}$. The expected, complete log-likelihood (with terms constant with respect to $\mathcal{C}$ deleted) can be written as

$$l(v) = \mathrm{tr}\left\{\Omega \mathrm{mat}(\mathcal{C})\left[\Psi \mathrm{mat}(\mathcal{C})^{\mathrm{T}} - 2\Phi^{\mathrm{T}}\right]\right\}, \quad (11)$$

where $\Omega = \mathrm{mat}(\hat{\mathcal{R}})^{-1}$, $\Psi = \sum_{n=1}^{N} \mathrm{E}(\mathrm{vec}\,\mathcal{Z}_n \mathrm{vec}\,\mathcal{Z}_n^{\mathrm{T}})$, and $\Phi = \sum_{n=1}^{N} \mathrm{vec}\,(\mathcal{X}_n)\mathrm{E}(\mathrm{vec}\,\mathcal{Z}_n)^{\mathrm{T}}$. Now let $k$ correspond to some $C_{ij}^{(m)}$ and let $\Delta_{ij} \in \mathbf{R}^{I_m \times J_m}$ be the indicator matrix that is one at the $(i,j)^{\mathrm{th}}$ entry and zero elsewhere. The gradient $\nabla l(v) \in \mathbf{R}^{\sum_m I_m J_m}$ is given elementwise by

$$\nabla l(v)_k = 2\mathrm{tr}\left\{\Omega \partial_{v_k} \mathrm{mat}(\mathcal{C})\left[\Psi \mathrm{mat}(\mathcal{C})^{\mathrm{T}} - \Phi^{\mathrm{T}}\right]\right\}, \quad (12)$$

where $\partial_{v_k} \mathrm{mat}(\mathcal{C}) = C^{(M)} \otimes \cdots \otimes \Delta_{ij} \otimes \cdots \otimes C^{(1)}$ [1]. If $m = M$, then we can exploit the sparsity of $\partial_{v_k} \mathrm{mat}(\mathcal{C})$ by computing the trace of the product of two submatrices each with $\prod_{n \neq M} I_n$ rows and $\prod_{n \neq M} J_n$ columns:

$$\nabla l(v)_k = 2\mathrm{tr}\left\{\left[C^{(M-1)} \otimes \cdots \otimes C^{(1)}\right]^{\mathrm{T}} \Lambda_{ij}\right\}, \quad (13)$$

where $\Lambda_{ij}$ is the submatrix of $\Omega\left[\mathrm{mat}(\mathcal{C})\Psi - \Phi\right]$ with row indices $(1, \ldots, \prod_{n \neq M} I_n)$ shifted by $\prod_{n \neq M} I_n(i-1)$ and column indices $(1, \ldots, \prod_{n \neq M} J_n)$ shifted by $\prod_{n \neq M} J_n(j-1)$. If $m \neq M$, then the ordering of the modes can be replaced by $1, \ldots, m-1, m+1, \ldots, M, m$ and the rows and columns of $\Omega\left[\mathrm{mat}(\mathcal{C})\Psi - \Phi\right]$ can be permuted accordingly. In other words, the original tensors $\mathcal{X}_n$ are "rotated" so that the $m^{\mathrm{th}}$ mode becomes the $M^{\mathrm{th}}$ mode.

The M-step for $\mathcal{A}$ can be computed in a manner analogous to that of $\mathcal{C}$ by replacing $I$ by $J$, replacing $\mathrm{mat}(\mathcal{C})$ by $\mathrm{mat}(\mathcal{A})$, and substituting $v = [\mathrm{vec}(A^{(1)})^{\mathrm{T}} \cdots \mathrm{vec}(A^{(M)})^{\mathrm{T}}]^{\mathrm{T}}$, $\Omega = \mathrm{mat}(\mathcal{Q})^{-1}$, $\Psi = \sum_{n=1}^{N-1} \mathrm{E}\left[\mathrm{vec}(\mathcal{Z}_n)\mathrm{vec}(\mathcal{Z}_n)^{\mathrm{T}}\right]$, and $\Phi = \sum_{n=1}^{N-1} \mathrm{E}\left[\mathrm{vec}(\mathcal{Z}_{n+1})\mathrm{vec}(\mathcal{Z}_n)^{\mathrm{T}}\right]$ into (11).

### 4.3 Special cases of the MLDS and their relationships to existing models

It is clear that the MLDS is exactly an LDS in the case $M = 1$. Certain constraints on the MLDS also lead to generalizations of factor analysis, probabilistic principal components analysis (PPCA), the CP decomposition, and the matrix factorization model of collaborative filtering (MF). Let $p = \prod_{m=1}^{M} I_m$ and $q = \prod_{m=1}^{M} J_m$. If $\mathcal{A} = 0$, $\mathcal{U}_0 = 0$, and $\mathcal{Q}_0 = \mathcal{Q}$, then the $\mathcal{X}_n$ of the MLDS become independent and identically distributed draws from the multilinear Gaussian distribution. Setting $\text{mat}(\mathcal{Q}) = \text{Id}_q$ and $\text{mat}(\mathcal{R})$ to a diagonal matrix results in a model that reduces to factor analysis in the case $M = 1$. A further constraint on $\mathcal{R}$, $\text{mat}(\mathcal{R}) = \rho^2 \text{Id}_p$, yields a multilinear extension of PPCA. Removing the constraints on $\mathcal{R}$ and forcing $\text{mat}(\mathcal{Z}_n) = \text{Id}_q$ for all $n$ results in a probabilistic CP decomposition in which the tensor elements have general covariances. Finally, the constraint $M = 2$ yields a probabilistic MF.

## 5 Experimental results

To determine how well the MLDS could model tensor time series, the fits of the MLDS were compared to those of the LDS for both synthetic and real data. To avoid unnecessary complexity and highlight the difference between the two models—namely, how the transition and projection operators are defined—the noises in the models are isotropic. The MLDS parameters are initialized so that $\mathcal{U}_0$ is drawn from the standard normal distribution, the matricizations of the covariance tensors are identity matrices, and the columns of each $A^{(m)}$ and $C^{(m)}$ are the first $J_m$ eigenvectors of singular-value-decomposed matrices with entries drawn from the standard normal distribution. The LDS parameters are initialized in the same way by setting $M = 1$.

The prediction error and convergence in likelihood were measured for each dataset. For the synthetic dataset, model complexity was also measured. The prediction error $\epsilon_n^{\mathcal{M}}$ of a given model $\mathcal{M}$ for the $n^{\text{th}}$ member of a tensor time series $\mathcal{X}_1, \ldots, \mathcal{X}_N$ is the relative Euclidean distance $||\mathcal{X}_n - \mathcal{X}_n^{\mathcal{M}}|| / ||\mathcal{X}_n||$, where $||\cdot|| = ||\text{vec}(\cdot)||_2$. Each estimate $\mathcal{X}_n^{\mathcal{M}}$ is given by $\mathcal{X}_n^{\mathcal{M}} = \text{vec}_I^{-1} \left( \text{mat}(\mathcal{C}^{\mathcal{M}}) \, \text{mat}(\mathcal{A}^{\mathcal{M}})^n \, \text{vec}(\text{E}[\mathcal{Z}_{N_{\text{train}}}^{\mathcal{M}}]) \right)$, where $\text{E}[\mathcal{Z}_{N_{\text{train}}}^{\mathcal{M}}]$ is the estimate of the latent state of the last member of the training sequence. The convergence in likelihood of each model is determined by monitoring the marginal likelihood as the number of EM iterations increases. Each model is allowed to run until the difference between consecutive log-likelihood values is less than 0.1% of the latter value. Lastly, the model complexity is determined by observing how the likelihood and prediction error of each model vary as the model size $|\theta^{\mathcal{M}}|$ increases. Aside from the model complexity experiment, the LDS latent dimensionality is always set to the smallest value such that the number of parameters of the LDS is greater than or equal to that of the MLDS.

### 5.1 Results for synthetic data

The synthetic dataset is an MLDS with dimensions $I = (7, 11)$, $J = (3, 5)$, and $N = 1100$ and parameters initialized as described in the first paragraph of this section. For the prediction error and convergence analyses, the latent dimensionality of the MLDS for fitting was set to $J = (3, 5)$ as well. Each model was trained on the first 1000 elements and tested on the last 100 elements of the sequence. The results are shown in Figure 3. According to Figure 3(a), the prediction error of MLDS matches that of the true model and is below that of the LDS. Furthermore, the MLDS converges to the likelihood of the true model, which is greater than that of the LDS (see Figure 3(b)). As for model complexity, the model size needed for the MLDS to match the likelihood and prediction error of the true model is much smaller than that of the LDS (see Figure 3(c) and (d)).

### 5.2 Results for real data

We consider the following datasets:

**SST:** A 5-by-6 grid of sea-surface temperatures from $5°$N, $180°$W to $5°$S, $110°$W recorded hourly from 7:00PM on 4/26/94 to 3:00AM on 7/19/94, yielding 2000 epochs [7].

**Tesla:** Opening, closing, high, low, and volume of the stock prices of 12 car and oil companies (e.g., Tesla Motors Inc.), from 6/29/10 to 5/10/13 (724 epochs).

**NASDAQ-100:** Opening, closing, adjusted-closing, high, low, and volume for 20 randomly-chosen NASDAQ-100 companies, from 1/1/05 to 12/31/09 (1259 epochs).

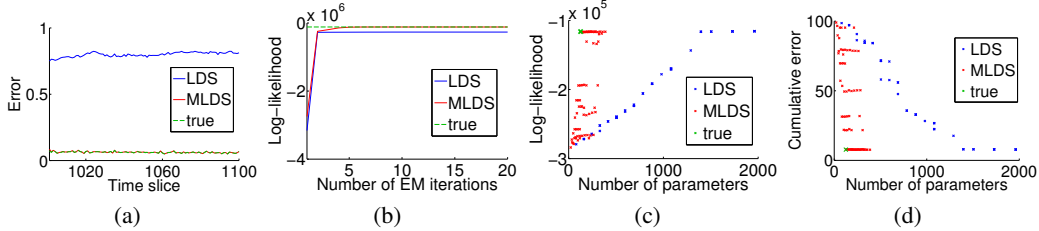

(a)  (b)  (c)  (d)

**Figure 3:** Results for synthetic data. Prediction error $\epsilon_n^{\mathcal{M}} = \left|\left| \mathcal{X}_n - \mathcal{X}_n^{\mathcal{M}} \right|\right| / \left|\left| \mathcal{X}_n \right|\right|$ is shown as a function of the time slice $n$ in (a), convergence of marginal log-likelihood is shown in (b), marginal log-likelihood as a function of model size is shown in (c), and cumulative prediction error $\sum_{n=N_{\text{train}}+1}^{N_{\text{train}}+N_{\text{test}}} \epsilon_n^{\mathcal{M}}$ as a function of model size is shown in (d) for LDS, MLDS, and the true model.

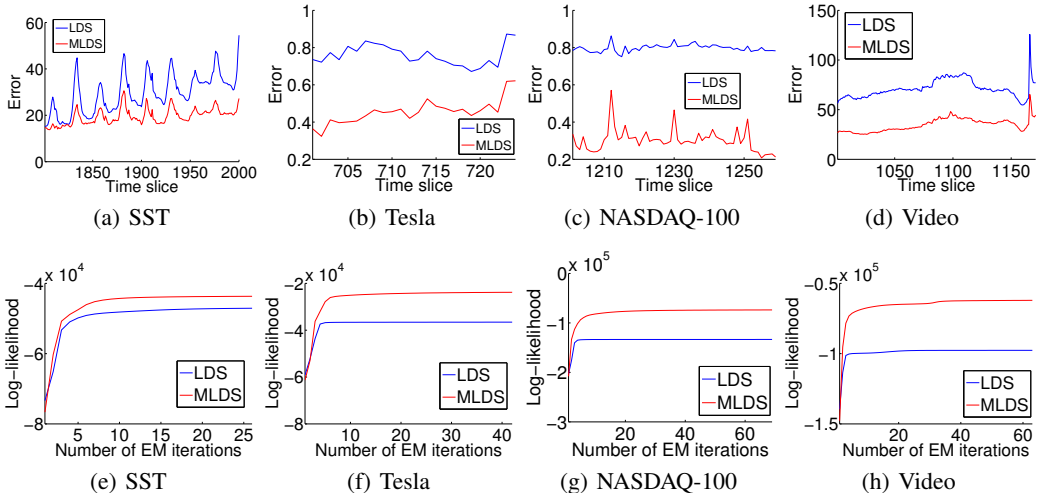

(a) SST  (b) Tesla  (c) NASDAQ-100  (d) Video

(e) SST  (f) Tesla  (g) NASDAQ-100  (h) Video

**Figure 4:** Results for LDS and MLDS applied to real data. The first row corresponds to prediction error $\epsilon_n^{\mathcal{M}}$ as a function of the time slice $n$, while the second corresponds to convergence in log-likelihood. Sea-surface temperature, Tesla, NASDAQ-100, and Video results are given by the respective columns.

**Video:** 1171 grayscale frames of ocean surf during low tide. This dataset was chosen because it records a quasiperiodic natural scene.

For each dataset, MLDS achieved higher prediction accuracy and likelihood than LDS. For the SST dataset, each model was trained on the first 1800 epochs; occlusions were filled in using linear interpolation and refined with an extra step during the learning that replaced the estimates of the occluded values by the conditional expectation given all the training data. For results when the MLDS dimensionality is set to $(3, 3)$, see Figure 4(a) and (e). For the Tesla dataset, each time series $((\mathcal{X}_1)_{ij}, \ldots, (\mathcal{X}_N)_{ij})$ were normalized prior to learning by subtracting by the mean and dividing by the standard deviation. Each model was trained on the first 700 epochs. See Figure 4(b) and (f) for results when the MLDS dimensionality is set to $(5, 2)$. For the NASDAQ-100 dataset, each model was trained on the first 1200 epochs. The data were normalized in the same way as with the Tesla dataset. For results when the MLDS dimensionality is set to $(10, 3)$, see Figure 4(c) and (g). For the Video dataset, a 100-by-100 patch was selected, spatially downsampled to a 10-by-10 patch for each frame, and normalized as before. Each model was trained on the first 1000 frames. See Figure 4(d) and (h) for results when the MLDS dimensionality is set to $(5, 5)$.

## 6 Related work

Several existing models can be fitted to tensor time series. If each tensor is "vectorized", i.e., reexpressed as a vector so that each element is indexed by a single positive integer, then an LDS can be applied [8, 6]. An obvious limitation of the LDS for modeling tensor time series is that the tensor structure is not preserved. Thus, it is less clear how the latent vector space of the LDS relates to the various tensor modes. Further, one cannot postulate a latent dimension for each mode as with the MLDS. The net result, as we have shown, is that the LDS requires more parameters than the MLDS to model a given system (assuming it does have tensor structure).

Dynamic tensor analysis (DTA) and Bayesian probabilistic tensor factorization (BPTF) are explicit models of tensor time series [9, 10]. For DTA, a latent, low-dimensional "core" tensor and a set of projection matrices are learned by processing each member $\mathcal{X}_n \in \mathbf{R}^{I_1 \times \cdots \times I_M}$ of the sequence as follows. For each mode $m$, the tensor is flattened into a matrix $\mathcal{X}_n^{(m)} \in \mathbf{R}^{(\prod_{k \neq m} I_k) \times I_m}$ and then multiplied by its transpose. The result $\mathcal{X}_n^{(m)\mathrm{T}} \mathcal{X}_n^{(m)}$ is added to a matrix $S^{(m)}$ that has accumulated the flattenings of the previous $n-1$ tensors. The eigenvalue decomposition $U \Lambda U^{\mathrm{T}}$ of the updated $S^{(m)}$ is then computed and the $m^{\mathrm{th}}$ projection matrix is given by the first $\mathrm{rank}\left(S^{(m)}\right)$ columns of $U$. After this procedure is carried out for each mode, the core tensor is updated via the multilinear transformation given by the Tucker decomposition. Like the LDS, DTA is a sequential model. An advantage of DTA over the LDS is that the tensor structure of the data is preserved. A disadvantage is that there is no straightforward way to predict future terms of the tensor time series. Another disadvantage is that there is no mechanism that allows for arbitrary noise relationships among the tensor elements. In other words, the noise in the system is assumed to be isotropic.

Other families of isotropic models have been devised that "tensorize" the time dimension by concatenating the tensors in the time series to yield a single new tensor with an additional temporal mode. These models include multilinear principal components analysis [11], the memory-efficient Tucker algorithm [12], and Bayesian tensor analysis [13]. For fitting to data, such models rely on alternating optimization methods, such as alternating least squares, which are applied to each mode.

BPTF allows for prediction and more general noise modeling than DTA. BPTF is a multilinear extension of collaborative filtering models [14, 15, 16] that concatenates the members of the tensor time series $(\mathcal{X}_n), \mathcal{X}_n \in \mathbf{R}^{I_1 \times \cdots \times I_M}$, to yield a higher-order tensor $\mathcal{R} \in \mathbf{R}^{I_1 \times \cdots \times I_M \times K}$, where $K$ is the sequence length. Each element of $\mathcal{R}$ is independently distributed as $\mathcal{R}_{i_1 \cdots i_M k} \sim \mathcal{N}(\langle u_{i_1}^{(1)}, \ldots, u_{i_M}^{(M)}, T_k \rangle, \alpha^{-1})$, where $\langle \cdot, \ldots, \cdot \rangle$ denotes the tensor inner product and $\alpha$ is a global precision parameter. Bayesian methods are then used to compute the canonical-decomposition/parallel-factors (CP) decomposition of $\mathcal{R}$: $\mathcal{R} = \sum_{r=1}^R u_r^{(1)} \circ \cdots \circ u_r^{(M)} \circ T_r$, where $\circ$ is the tensor outer product. Each $u_r^{(m)}$ is independently drawn from a normal distribution with expectation $\mu_m$ and precision matrix $\Lambda_m$, while each $T_r$ is recursively drawn from a normal distribution with expectation $T_{r-1}$ and precision matrix $\Lambda_T$. The parameters, in turn, have conjugate prior distributions whose posterior distributions are sampled via Markov-chain Monte Carlo for model fitting. Though BPTF supports prediction and general noise models, the latent tensor structure is limited.

Other models with anisotropic noise include probabilistic tensor factorization (PTF) [17], tensor probabilistic independent component analysis (TPICA) [18], and generalized coupled tensor factorization (GCTF) [19]. As with BPTF, PTF and TPICA utilize the CP decomposition of tensors. PTF is fit to tensor data by minimizing a heuristic loss function that is expressed as a sum of tensor inner products. TPICA iteratively flattens the tensor of data, executes a matrix model called probabilistic ICA (PICA) as a subroutine, and decouples the factor matrices of the CP decomposition that are embedded in the "mixing matrix" of PICA. GCTF relates a collection of tensors by a hidden layer of disconnected tensors via tensor inner products, drawing analogies to probabilistic graphical models.

## 7  Conclusion

We have proposed a novel probabilistic model of tensor time series called the multilinear dynamical system (MLDS), based on a tensor normal distribution. By putting tensors and multilinear operators in bijective correspondence with vectors and matrices in a way that preserves tensor structure, the MLDS is formulated so that it becomes an LDS when its components are vectorized and matricized. In matrix form, the transition and projection tensors can each be written as the Kronecker product of $M$ smaller matrices and thus yield an exponential reduction in model complexity compared to the unfactorized transition and projection matrices of the LDS. As noted in Section 4.3, the MLDS generalizes the LDS, factor analysis, PPCA, the CP decomposition, and low-rank matrix factorization. The results of multiple experiments that assess prediction accuracy, convergence in likelihood, and model complexity suggest that the MLDS achieves a better fit than the LDS on both synthetic and real datasets, given that the LDS has the same number of parameters as the MLDS.

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
