[Reviews · NeurIPS 2013]

Submitted by Assigned_Reviewer_3

The wealth of multi-dimensional structured data found in many applications requires methods for decomposition, modeling and analysis. This paper considers time-evolving multi-d arrays and considers generalizations of systems for vector/matrix dynamics. The paper is fairly well-written and offers an exposition of basic tensor manipulations - I checked the math and it appears sound.

I have some comments and qns as below:
1) why are these multi-d objects tensors? A multi-d array is in general not a tensor, just a structured collection of indexed data. A tensor has more fundamental properties - of invariance to coordinates for example, as a tensor is a *geometric object* that defines relationships between vectors and matrices in generic coordinates? Perhaps the confusion lies in the common usage of "tensor" to describe the scalar coefficients of the tensor, which are indeed just scalars in a multi-d array. I am sure I'm not alone in being confused by understanding how an array of financial timeseries can be regarded as a tensor?

2) There is considerable work in the fmri literature on dynamics, factorizations and [scalar] tensor coefficients. How does your work relate to eg that of Beckmann et al?

3) Given that the tensor coefficients may be reshaped using generalized vec() operators, and that tensor structure may be preserved using structured indicator matrices - and so the system solved using standard dynamical modeling, what is the added benefit to direct tensor modeling?

4) parameter estimation in tensor, or high-d structure modeling is notoriously sensitive to priors, algorithm, latent dimension and amounts of data. I found little comment regarding your experiences.

5) The choice of latent decomposition is critically dependent upon the choice of the MLDS dimensionality. The paper quotes 'set' values. What is the sensitivity of the forecast error upon these?
Summary: An interesting topic, with some incremental theoretical work. The paper suffers from a potential lack of detail with respect to critical parameter settings and knock on sensitivity analysis. Some of the concepts are obscured to the reader not familiar to tensors, especially given the confused definition of a "tensor".

Submitted by Assigned_Reviewer_4

In this paper the authors propose a natural extension of linear dynamical systems to deal with sequences of tensors. There are two advantages: first, the information conveyed in the structure of the tensors is kept; second, the tensor factorization of the transition and projection operators allow for keeping a low number of learning parameters. The paper is well written and the idea developed is novel as far as I am concerned.

In the following I mention small details:
1.- I understand that tensor A in Sec. 5.1 has dimensionality J_1 x ... x J_M x J_1 x ... x J_M. This should be clarified because the previous time tensor A appears in the paper (in Lemma 1) it has different dimensionality.
2.- \Gamma appears in the statement of Lemma 2 but is only defined later in its proof.
3.- In the experiments the number of tensors in a sequence is denoted by N, whereas before it is used K.

In the experiments it is shown the superiority of the proposed method with respect to LDS, as one may expect a priori. It would also be interesting to show the running time (according to the description of the algorithm, it looks quite demanding).

I acknowledge that I have read the author rebuttal and the reviews. I think the paper is acceptable, although after reading the reviews (Stephen, especially point 3) and rebuttal, I agree I have overestimated a bit the paper in my first appraisal and I will decrease it.
Summary: Interesting extension of linear dynamical systems to consider tensor sequences. Novel method and well written paper.

Submitted by Assigned_Reviewer_5

This paper proposes a tensorial LDS model called MLDS. As such, the model is a restriction of the general Kalman filter model with state transition and observation operators having a certain Kronecker product factorization. The model can be understood intuitively as a sequence of core tensors that form a Markov Chain, and the observed tensor at each time slice as well as the core tensor at the next time slice are obtained by Tucker models with tensor factors C and A respectively. The term multilinear in the title is misleading in this respect as the model is linear.

After setting up the notation and background, authors derive the posterior distribution of hidden tensors given the observed tensor in Sec. 4, Lemma 2. This is direct generalization of standard marginalization and conditioning results of multivariate Gaussians. What is not discussed is the form of the state covariance and whether it admits a useful factorization (compare with matrix Gaussians where the covariance is given as a Kronecker Product). I suspect that this model would not admit such a structured covariance but I would have welcomed at least a discussion of this issue because calculating the covariance would be the computational bottleneck with large Z, not much mentioned here and somewhat swept under the carpet.

The ideas are quite analogous to theory of linear dynamical systems (LDS) but the tensor notation is unevitably quite heavy, yet the authors made a pretty good job in keeping the notation intuitive by drawing analogies with the vector case. Perhaps not surprisingly, the paper concludes that by preserving the linear structure, MLDS is more effective than a plain LDS where the tensors are simply vectorized. However, the paper does not much detail on inference (filtering/smoothing recursions) and instead focuses on parameter inference via EM.


Another issue I would have seen discussed is identifiability and the justification of the Tucker transition model. The LDS states are identifiable upto similarity transform, what is the case for the case discussed here? For the latter point, whilst the observation model is clear, it would be good at least to justify the Tucker transition model; does this have a physical meaning? Otherwise I would suspect taking an unconstrained transition operator A or the other extreme of taking A as the identity (e.g., a drifting core tensor) are equally viable.


Sec. 5.2 develops an EM algorithm to for parameter estimation. This section is rather short and many details are omitted. The classical EM algorithm for learning an LDS derives update rules for each parameter, including noise covariances. In the paper, although authors state that, ‘MLDS inherits the E-step of the LDS and the M-step of the LDS for all parameters except A and C’, it is not clear what they mean and it is not possible to see the update rules of noise covariances at one-step. Furthermore, authors do not explicitly state the update rules for A and C and merely derive the gradient. In the Eq. (11), authors use \phi and \varphi which are defined with expectations. In the LDS, these expectations require to find smoothed estimates (in addition to filtered estimates). The paper nowhere mentions the smoothing recursions explicitly and the reader has to simply redrive the results from scratch. From the description in the paper it is not clear that how these expectations are computed making the work hard to reproduce.

Despite all the small glitches, I find the model interesting and well executed enough deserving publication.
I am surprised that the literature does not contain special cases of this model such as 'Matrix variate Kalman Filter'.


Short comments
Abstract and introduction should be more clear and state the precise results about inference and learning algorithms that are obtained in this paper.

Experimental section is good. If authors think to resubmit this paper, they should focus on the Section 4 and Section 5.

The statement in the related work section page 3, line 122 is not entirely correct as GCTF is not limited to CP but can accomodate arbitrary factorization forms and fairly general noise models.

The statement in the conclusion "permitting us to learn
all but the transition and projection operators A and C of the MLDS via the Kalman filter"
is vague (and actually incorrect as KF provides the filtered state estimates and covariances)

The citation to [18] looks off-topic and can either be omitted or must be elaborated more to stress why the non existence may be an issue.
Summary: A potentially useful and elegant special case of the LDS for tensor variate data is developed. The results would be hard to reproduce as details about inference are missing.

Submitted by Assigned_Reviewer_6

This paper presents a formulation for linear dynamic systems when the observed and latent variables are tensors. The key is to use the tensor-variate normal distribution in place of the standard vector-variate normal. The authors show that one can handle the tensor-normal, using a simple algebra between tensors that is quite similar to the standard vector/matrix algebra. They extend the standard EM approach for model inference of LDS to tensor variables by taking advantage of the algebra they derived. Finally, they compared the model with vectorized LDS for matrix time-series (M=2) when the dimensionality of the latent tensor is known.

This paper is well-written. The authors give generic and useful theorems for tensor algebra useful in handling tensor-variate normal. To the best of my knowledge (I'm not extremely familiar with the domain, though), this is the first paper that derives an EM algorithm based on the tensor-variate normal. I think we should recognize the author for the effort.

The extension of LDS to tensors is interesting enough from a theoretical perspective, but I wonder if such an extension is really rewarding. In fact, the authors only give M=2 examples for given latent dimensionalities. In that case, the tensor algebra is reduced to the standard matrix algebra as described in standard multivariate analysis textbooks. In a journal version of the paper, I'd like to see simplified formulas customized for the M=2 case, where all the mysteriously-looking symbols are not necessary.
Summary: This paper appears to be the first work deriving an EM algorithm for tensor-variate LDS based on a systematic treatment of tensor-variate Gaussian. It should be accepted if practical utility is rather questionable for M larger than 2.
Author Feedback

Author rebuttal: Response to Assigned_Reviewer_3
Thank you for the constructive feedback.

Our definition of a tensor as a multidimensional array is consistent with the definition in [2].

Thanks to your suggestion, we will discuss TPICA of "Tensorial extensions of independent component analysis for multisubject FMRI analysis" by Beckmann et al. (2005). The MLDS is composed of Tucker operators while TPICA belongs to the family of CP decompositions. Note that the CP decomposition is a special case of the Tucker decomposition.

We agree that a structured indicator matrix formulation is possible but believe it may lead to a more complicated representation of the same model. Our direct tensor modeling offers a great reduction in model complexity for tensor time series and the formulation is simple and intuitive.

When the tensors are vectorized, our model becomes a constrained LDS; our experiments focused solely around the consequences of those constraints. The number of parameters is determined by the number of MLDS dimensions. The forecast error is shown as a function of the number of parameters in Figure 3(d). To illustrate the forecast error as the number of dimensions of each mode increases requires a graph with M+1 axes, where M is the number of modes. Such a graph cannot be visualized in general.

Response to Assigned_Reviewer_4
Thank you for pointing out the errata. We will fix all of them.

Each iteration for maximizing (11) takes O(p^M) time, where p is the number of elements of the largest projection matrix and M is the number of modes.

Response to Assigned_Reviewer_5
Thank you for the helpful comments. We will correct the GCTF statement. “Multilinear” in the title refers to the multilinear Tucker operators of the MLDS, which can be found in "Multilinear operators for higher-order decompositions" by Kolda (2006). We show that the multilinear operators of the MLDS can be represented as linear operators via matricization.

Note that when the tensors are vectorized, the MLDS becomes a LDS with additional constraints on the transition and projection operators. The smoothing equations for inference of the MLDS latent states are almost the same as those for LDS. Due to the space limit, we omited these details. We will make this more clear in the revision so that the reader understands that there is no need to rederive anything. We will offer a longer technical report version for reproduction. The matricized covariances of the MLDS are computed in the similar way as the covariances of LDS are. On this note, we found in the experiments that the main bottleneck in the computation lies in computing A and C and not in computing covariances.

Although the discussion of identifiability is beyond the scope of our work, we believe the MLDS is also identifiable up to similarity transform.

Response to Assigned_Reviewer_6
Thank you for the discussion. Because tensor methods such as ours are required for M>1, we chose M=2 examples for simplicity. Many M>2 examples exist; we discussed one of them in the introduction. Nevertheless, we will simplify the notation for the M=2 case.